# PyTorch-Geometric Edge – a Library for Learning Representations of Graph Edges

**Piotr Bielak** and **Tomasz Kajdanowicz**

Department of Artificial Intelligence,
Wrocław University of Science and Technology,
Wrocław, Poland
`piotr.bielak@pwr.edu.pl`

## Abstract

Machine learning on graphs (GraphML) has been successfully deployed in a wide variety of problem areas, as many real-world datasets are inherently relational. However, both research and industrial applications require a solid, robust, and well-designed code base. In recent years, frameworks and libraries, such as PyTorch-Geometric (PyG) or Deep Graph Library (DGL), have been developed and become first-choice solutions for implementing and evaluating GraphML models. These frameworks are designed so that one can solve any graph-related task, including node- and graph-centric approaches (e.g., node classification, graph regression). However, there are no edge-centric models implemented, and edge-based tasks are often limited to link prediction. In this extended abstract, we introduce `PyTorch-Geometric Edge` (PyGE), a deep learning library that focuses on models for learning vector representations of edges. As the name suggests, it is built upon the PyG library and implements edge-oriented ML models, including simple baselines and graph neural networks, as well as corresponding datasets, data transformations, and evaluation mechanisms. The main goal of the presented library is to make edge representation learning more accessible for both researchers and industrial applications, simultaneously accelerating the development of the aforementioned methods, datasets and benchmarks.

## 1 Introduction

Nowadays, one of the most prominent research areas in machine learning is representation learning. Solving classification, regression, or clustering tasks by means of popular machine learning models, like decision trees, SVMs, logistic regression, linear regression, or feed-forward neural networks, requires the presence of object features in the form of real-valued number vectors (also called embeddings, or representation vectors). Representation learning aims at finding algorithms and models that can extract such numeric features from arbitrary objects (images, texts, or graphs) in an automated and reliable way. In terms of machine learning on graphs (GraphML), these models / algorithms are called *graph representation learning (GRL)* methods. In recent years, GRL methods have been successfully deployed in a wide variety of domains, including social networks, financial networks, and computational chemistry [1–4].

This wide adoption of graph-based models led to the creation of publicly available implementations, often in the form of frameworks or libraries with standardized APIs, which describe data formats, model building blocks, and scalable parameter optimization techniques. First-choice solutions are currently frameworks like PyTorch-Geometric (PyG) [5] or the Deep Graph Library (DGL) [6]. They include most of the existing graph neural networks and some traditional models, as well as datasets, preprocessing transformations, and basic evaluation mechanisms. This simplifies both production-ready model development and conducting GraphML research.

P. Bielak, T. Kajdanowicz, PyTorch-Geometric Edge – a Library for Learning Representations of Graph Edges (Extended Abstract). Presented at the First Learning on Graphs Conference (LoG 2022), Virtual Event, December 9–12, 2022.

The implemented design choices allow solving any graph-related task (e.g., node classification, graph regression). Nevertheless, the main focus in these libraries is on node- and graph-centric models and tasks, whereas edge-based tasks are often limited to link prediction.

**Present work.** We aim to fill the gap for edge-centric GRL models and tasks. In this extended abstract, we introduce `PyTorch-Geometric Edge` (PyGE), a deep learning library focused on models for learning vector representations of graph edges. We build upon the PyTorch-Geometric (PyG) library and provide implementations: (1) for edge-centric models, including simple baselines and graph neural networks, (2) edge-based GNN layers, (3) datasets and corresponding preprocessing functions (in a PyTorch- and PyG-compliant format), and (4) evaluation mechanisms for edge tasks. PyGE should make edge representation learning more accessible for both researchers and industrial applications, simultaneously accelerating the development of edge-centric methods, datasets and benchmarks. **Disclaimer:** Please note that the introduced library is still under active development. We provide a summary of our planned work in Section 4.

**Contributions.** We summarize our contributions as follows: (C1) We publicly release `PyTorch-Geometric Edge`, the first deep learning library for edge representation learning – https://github.com/pbielak/pytorch_geometric_edge. (C2) We implement a subset of available edge-based models, graph neural network layers, datasets, and corresponding data transformations.

## 2 Preliminaries

We start by introducing definitions for basic concepts covered in our presented library and explore the current state of node and edge embedding approaches, as well as GraphML software.

**Graph.** A graph $\mathcal{G} = (\mathcal{V}, \mathcal{E})$ describes a set of nodes $\mathcal{V}$ that are connected (pairwise) by a set of edges $\mathcal{E} \in \mathcal{V} \times \mathcal{V}$. An **attributed** graph $\mathcal{G} = (\mathcal{V}, \mathcal{E}, \mathbf{X}, \mathbf{X}^{\text{edge}})$ extends this definition by a set of node attributes: $\mathbf{X} \in \mathbb{R}^{|\mathcal{V}| \times d_{\text{node}}}$, and optionally also edge attributes: $\mathbf{X}^{\text{edge}} \in \mathbb{R}^{|\mathcal{E}| \times d_{\text{edge}}}$.

**Edge representation learning.** The goal is to find a function $f_\theta : \mathcal{E} \to \mathbb{R}^{d_{\text{edge}}}$ that maps an edge $e_{(u,v)} \in \mathcal{E}$ into a low-dimensional ($d_{\text{edge}} \ll \dim(\mathcal{E})$) vector representation (embedding) $\mathbf{z}_{uv}$ that preserves selected properties of the edge (e.g., features or local structural neighborhood information).

**Edge-based tasks.** Evaluation tasks for edge embeddings include: (1) link prediction – binary classification problem of the existence (future appearance) of an edge; (2) edge classification – label/type prediction of an existing edge (e.g., kind of social network relation); (3) edge regression – prediction a numerical edge feature (e.g., bond strength in a molecule).

**Node representation learning methods.** Early approaches were built around the transductive setting with an enormous **trainable lookup-embedding matrix**, whose rows denote representation vectors for each node. The optimization process would preserve structural node information. For instance, DeepWalk [7], and its successor Node2vec [8] use the Skipgram [9] objective to model random walk-based co-occurrence probabilities. TADW [10] extended this approach to attributed graphs and reformulated the model as a matrix factorization problem. Other early approaches include: LINE [11], SDNE [12], or FSCNMF [13]. Recent methods are based on **Graph Neural Networks** (GNNs) – trainable functions that transform feature vectors of a node and its neighbors to a new embedding vector (inductive setting). These functions can be stacked to create a deep (graph) neural network. The most popular ideas include: a graph reformulation of the convolution operator (GCN [14]), neighborhood sampling and aggregation of sampled features (GraphSAGE [15]), attention mechanism over graph structure (GAT [16]) or modeling injective functions (GIN [17]).

**Edge representation learning methods.** This area is still underdeveloped, i.e., only a handful of proposed models and algorithms exists. Most early approaches are **node-based transformations**, i.e., the edge embedding $\mathbf{z}_{uv}$ is computed from two node embeddings $\mathbf{z}_u$ and $\mathbf{z}_v$. There are simple **non-trainable binary operators** [8], such as the average ($\mathbf{z}_{uv} = \frac{\mathbf{z}_u + \mathbf{z}_v}{2}$), the Hadamard product ($\mathbf{z}_{uv} = \mathbf{z}_u * \mathbf{z}_v$), or the weighted L1 ($\mathbf{z}_{uv} = |\mathbf{z}_u - \mathbf{z}_v|$) or L2 ($\mathbf{z}_{uv} = |\mathbf{z}_u - \mathbf{z}_v|^2$) operators. NRIM [18] proposes trainable transformations as two kinds of neural network layers: **node2edge**

($\mathbf{z}_{uv} = f_\theta([\mathbf{z}_u, \mathbf{z}_v, \mathbf{x}_{uv}^{\text{edge}}])$) and **edge2node** ($\mathbf{z}_u = f_\omega([\sum_{v \in \mathcal{N}(u)} \mathbf{z}_{uv}, \mathbf{x}_u])$). Another group of edge embedding methods **directly** learn the edge embeddings, i.e., without an intermediate node embedding step. **Line2vec** [19] utilizes a line graph transformation (converting nodes into edges and vice versa), applies a custom edge weighting method and runs Node2vec on the line graph. The loss function extends the Skipgram loss with a so-called *collective homophily* loss (to ensure closeness of neighboring edges in the embedding space). This method is inherently transductive (due to Node2vec) and completely ignores any attributes. Those problems are addressed by **AttrE2vec** [20]. It samples a fixed number of uniform random walks from two edge neighborhoods ($\mathcal{N}(u)$, $\mathcal{N}(v)$) and aggregates feature vectors of encountered edges (using average, exponential decaying, or recurrent neural networks) into summary vectors $\mathbf{S}_u$, $\mathbf{S}_v$, respectively. An MLP encoder network with a self-attention-like mechanism transforms the summary vectors and the edge features into the final edge embedding. AttrE2vec is trained using a contrastive cosine learning objective and a feature reconstruction loss. **PairE** [21] utilizes two kinds of edge feature aggregations: (1) concatenated node features (*self features*), (2) concatenation of averaged neighbor features for both nodes (*agg features*). An MLP encoder with skip-connections transforms these two vectors into the edge embedding. Two shallow decoders reconstruct the feature probability distribution. The resulting PairE autoencoder is trained using the sum of the KL-divergences of the *self* and *agg* features. **EHGNN** [22] proposes a so-called Dual Hypergraph Transformation (DHT) that inverts the role of nodes and edges – similarly to the line graph transformation, but with a lower time complexity. DHT can be paired with any existing node-based GNN approach to obtain the final edge embeddings. Other methods include: EGNN [23], ConPI [24] or Edge2vec [25].

**GraphML software.** The backbone of all modern deep learning frameworks are tools for automatic differentiation, such as: Tensorflow [26] or PyTorch [27]. GraphML libraries are mostly built upon these tools, e.g., PyG uses PyTorch, GEM [28] and DynGEM [29] use Tensorflow, DGL can be used both with Tensorflow and PyTorch, whereas some like KarateClub [30] are using a custom backend. All of these libraries are focused on node- and graph-centric models. Our proposed `PyTorch-Geometric Edge` library is the first one that focuses on edge-centric models and layers. It adapts the PyG library API and uses PyTorch as its backend.

## 3  PyTorch-Geometric Edge

**Relation to PyG.** Our proposed PyGE library re-uses the API and data format implemented in PyTorch-Geometric. The graph is stored as a `Data()` object with edges in form of a sparse COO matrix (`edge_index`). Other fields include: `x` (node attributes), `edge_attr` (edge attributes), `y` (node/edge labels). We also keep a similar layout of the library package structure, i.e., we have a module for datasets, models, neural network layers (`nn`), data transformations (`transforms`) and data samplers (`samplers`). The `forward()` method in all implemented models/layers accepts two parameters: `x` (node or edge features) and `edge_index` (adjacency matrix). Hence, the implemented models/layers can be integrated with other PyG models/layers and vice versa (we show that in the `examples/` folder in the repository). The same applies for the datasets.

### 3.1  Current state of implementation

We now show the current state of the library and what is already implemented. Please refer to Section 4 where we explain our future plans.

**Datasets.** We currently include 5 datasets (Cora, PubMed, KarateClub, Dolphin and Cuneiform) that were originally used in the papers of the implemented methods. We summarize their statistic in Table 1. Note most of them also require preprocessing steps (see: AttrE2vec [20] for details) for the edge classification evaluation – we implement appropriate data transformations. Moreover, we add cybersecurity-based datasets – UNSW-NB15 in four different versions. These datasets can be directly used for edge classification. We create different versions of this dataset by using: (1) different definitions of nodes (either just the IP address – yielding 49 different nodes, or using a combination of both the IP address and the port – about 1.1M nodes); (2) different class labels (either binary classification: attack/normal traffic – two classes, or using a more fine-grained attack definition – 14 classes). The number of edges corresponds to the number of connections – about 2.5M. Note that the number of features is higher than the one reported in the original paper [31] (49 features) – for the

categorical ones, like *protocol*, *state* or *serivce*, we already applied a one-hot encoding (yielding 202 or 204 features in total).

**Table 1:** Summary of included datasets. The ∗ symbol denotes the number of edge classes after applying an appropriate data transformation.

| Name | $|\mathcal{V}|$ | $|\mathcal{E}|$ | $d_{\text{node}}$ | $d_{\text{edge}}$ | classes |
|------|------|------|------|------|------|
| KarateClub [32] | 34 | 156 | - | - | 4* |
| Dolphin [33] | 62 | 318 | - | - | 5* |
| Cora [34] | 2 708 | 10 556 | 1 433 | - | 8* |
| PubMed [35] | 19 717 | 88 648 | 500 | - | 4* |
| Cuneiform [36] | 5 680 | 23 922 | 3 | 2 | 2 |
| UNSW-NB15 (IP) [31] | 49 | 2 539 739 | - | 204 | 2 / 14 |
| UNSW-NB15 (IP-Port) [31] | 1 112 275 | 2 539 739 | - | 202 | 2 / 14 |

**Models and layers.** We implement most of the edge representation learning methods discussed in Section 2 into our proposed PyGE library (see: Table 2). Nevertheless, more of them will be implemented in future versions.

**Table 2:** Models and layers implemented in PyGE.

| Method | Type | Inductive | Attributed | Characteristics |
|--------|------|-----------|------------|-----------------|
| Node pair op [8] | layer | ✓ | ✗ | non-trainable |
| node2edge [18] | layer | ✓ | ✓ | trainable |
| Line2vec [19] | model | ✗ | ✗ | line graph, random-walk |
| AttrE2vec [20] | model | ✓ | ✓ | contrastive, AE, random-walk |
| PairE [21] | model | ✓ | ✓ | AE, KL-div |
| EHGNN [22] | framework | ✓ | ✓ | time efficient, hypergraph |

**Embedding evaluation.** We implement a ready-to-use edge classification evaluator class, which takes edge embeddings and edge labels, applies a logistic regression classifier and returns typical classification metrics, like ROC-AUC, F1 or accuracy. This is a widely adopted technique in unsupervised learning, called the *linear evaluation protocol* [37].

**Example usage.** In the repository, we provide an end-to-end script showing the usage of a given model/layer. Every script: (1) loads a dataset and applies the required data transformations (preprocessing), (2) prepares the data split of edges into train and test sets, (3) builds a model, (4) trains the model for a certain amount of epochs, (5) evaluates the learned edge embeddings. We provide also an example script in this extended abstract – see Section A.

## 3.2 Maintenance

An open-source library requires continuous maintenance. We host our code base at GitHub, which allows to track all development progress and user-generated issues. We will build library releases and announce them on GitHub and host them later on the Python Package Index (PyPI) to allow users to simply run a `pip install torch-geometric-edge` command to install our library. We use the MIT license to give potential users, researchers, and industrial adopters a good user experience without worrying about the rights to use or modify our code base. Another aspect of software development and maintenance is Continuous Integration. We use the GitHub Actions module to automatically execute code quality checks and unit tests with every pull request to our library. This prevents that a change will break existing functionality or lower our assumed code quality.

## 4 Summary and roadmap

In this extended abstract, we presented an initial version of `PyTorch-Geometric Edge`, the first deep learning library that focuses on representation learning for graph edges. We provided information about currently implemented models/layers and datasets. Our roadmap is extensive and includes: (I) preparation of a complete documentation (right now: we rely on code quality checks and example scripts on how to use particular models/layers), (II) addition of more datasets (e.g., Enron Email Dataset[1], FF-TW-YT[2], among others), (III) implementation of other mentioned edge-centric models (and a continuous extension of the literature review to find new methods), (IV) we want to add more edge evaluation schemes, (V) in the full paper, we want to include an extensive benchmark of all implemented models and compare them in different downstream tasks; moreover we want to provide the entire reproducible experimental pipeline and pretrained models. With such an amount of incoming work, we want to encourage readers interested in edge representation learning to contact the authors and contribute to our library. We are convinced that edge representation learning can be widely adopted in networked tasks, like message classification in social networks, connection/attack classification in cybersecurity applications, to name only a few.

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

## A    Example 1: PairE model

Let's explore how to use PyGE in practice. We will be using the PairE model to classify the citation type between academic papers (citation within a research area or cross citation; if the same research area, then which one). We start by loading the Cora dataset and extracting the target edge labels using our implemented `MatchingNodeLabelsTransform()` (if two node labels match, use this label, else use special label $-1$):

```
from torch_geometric_edge.datasets import Cora
from torch_geometric_edge.transforms import MatchingNodeLabelsTransform

data = Cora("/tmp/pyge/", transform=MatchingNodeLabelsTransform())[0]
```

Next, we split the edges into train and test sets:

```python
import torch
from sklearn.model_selection import train_test_split

train_mask, test_mask = train_test_split(
    torch.arange(data.num_edges),
    stratify=data.y,
    test_size=0.8,
)
```

Now, let's create the PairE model:

```python
from torch_geometric_edge.models import PairE

model = PairE(
    num_nodes=data.num_nodes,
    node_feature_dim=data.num_node_features,
    emb_dim=128,
)
```

We can train our model using standard PyTorch training-loop boilerplate code. Note, that we only use training edges (`data.edge_index[:, train_mask]`).

```python
optimizer = torch.optim.AdamW(model.parameters(), lr=1e-3)

model.train()
for _ in range(100):
    optimizer.zero_grad()

    x_self, x_aggr = model.extract_self_aggr(data.x, data.edge_index[:, train_mask])
    h_edge = model(data.x, data.edge_index[:, train_mask])
    x_self_rec, x_aggr_rec = model.decode(h_edge)

    loss = model.loss(x_self, x_aggr, x_self_rec, x_aggr_rec)

    loss.backward()
    optimizer.step()
```

Finally, we can evaluate our model's edge embedding in the edge classification task using the `LogisticRegressionEvaluator`. The returned metrics will be prefixed to indicate the train/test split. Note that we use now all edges during inference:

```python
from torch_geometric_edge.evaluation import LogisticRegressionEvaluator

model.eval()
with torch.no_grad():
    Z = model(data.x, data.edge_index)

    metrics = LogisticRegressionEvaluator(["auc"]).evaluate(
        Z=Z,
        Y=data.y,
        train_mask=train_mask,
        test_mask=test_mask,
    )
print(metrics)
```

## B  Example 2: Node2Edge, Edge2Node layers

Let's explore another PyGE example code. We will be using the Node2Edge and Edge2Node layers to classify network traffic. We start by loading the UNSW-NB15 dataset:

```python
from torch_geometric_edge.datasets import UNSW_NB15

data = UNSW_NB15(version="ip/multi", root="/tmp/pyge/")[0]
```

Next, we split edge into train and test sets:

```python
import torch
from sklearn.model_selection import train_test_split

train_mask, test_mask = train_test_split(
    torch.arange(data.num_edges),
    stratify=data.y,
    test_size=0.8,
)
```

Now, we build a supervised model using the Node2Edge and Edge2Node layers:

```python
from torch import nn
from torch_geometric_edge.nn import Edge2Node, Node2Edge

class Model(nn.Module):

    def __init__(self, num_nodes: int, edge_dim: int, num_classes: int):
        super().__init__()
        self.e2n = Edge2Node(
            num_nodes=num_nodes,
            node_dim=0,
            edge_dim=edge_dim,
            out_dim=128,
        )
        self.n2e = Node2Edge(
            node_dim=128,
            edge_dim=edge_dim,
            out_dim=num_classes,
            net=nn.Sequential(
                nn.Linear(2 * 128 + edge_dim, 128),
                nn.ReLU(),
                nn.Linear(128, num_classes),
                nn.LogSoftmax(dim=-1),
            ),
        )

    def forward(
        self,
        edge_attr: torch.Tensor,
        edge_index: torch.Tensor,
    ) -> torch.Tensor:
        h = self.e2n(edge_attr=edge_attr, edge_index=edge_index)
        y_pred = self.n2e(x=h, edge_attr=edge_attr, edge_index=edge_index)

        return y_pred

model = Model(
    num_nodes=data.num_nodes,
    edge_dim=data.num_edge_features,
    num_classes=data.y.unique().shape[0],
)
```

Similarly to the previous example we build the train loop (using standard PyTorch boilerplate code) and evaluate our classifier:

```python
from sklearn.metrics import roc_auc_score
from torch.nn import functional as F

optimizer = torch.optim.AdamW(model.parameters(), lr=1e-3)

for _ in range(5):
    # Train
    model.train()
    optimizer.zero_grad()

    y_pred = model(data.edge_attr[train_mask], data.edge_index[:, train_mask])
    y_true = data.y[train_mask]

    loss = F.nll_loss(input=y_pred, target=y_true)
    print(loss)

    loss.backward()
    optimizer.step()

    # Evaluate
    model.eval()
    with torch.no_grad():
        y_score = model(data.edge_attr[test_mask], data.edge_index[:, test_mask]).exp()
        y_true = data.y[test_mask]

        test_auc = roc_auc_score(y_true=y_true, y_score=y_score, multi_class="ovr")
        print("Test AUC:", test_auc)
```

