# OpenReview forum: "PyTorch-Geometric Edge - a Library for Learning Representations of Graph Edges"
_logconference.io/LOG/2022/Conference — LoG 2022 Poster_

### Official Review · Reviewer_n7HR · 2022-10-17

**Overall Score:** 8
**Confidence:** 4

**Review:**

## Summary
This paper introduces the new library for edge representation learning for graph-structured data, which is based on well-known PyTorch and PyTorch-Geometric (PyG) libraries, yet dedicated to the edge-level graph representation learning. This paper describes the existing edge representation learning methods, as well as the datasets and the evaluation metrics for them, which are or would be implemented in the open-source code format.

---

## Strengths and Weaknesses

### Strengths
* The focus on edge representation learning is interesting, and the edge-level graph ML library is highly valuable for the graph ML community.
* The open-source-based maintenance plan with the MIT license: a right to use and modify the code base for people in both academia and industry, is good.
* This paper is generally well organized, and well written.

### Weaknesses
* In terms of edge-level tasks, the direction of edges is an important component to consider; however, it is not discussed in the paper. I am wondering this new library can support the direction of edges, and if not, it would be a clear weakness.
* There is no justification of releasing the code base with the new repository, not in the existing PyTorch Geometric repository. In my point of view, releasing the code under the popular PyTorch Geometric (i.e., PyG) repository is more valuable, since researchers can more easily access the edge-level functions/models without downloading the additional library; also, it might be easier to maintain the code under the PyG since many functions are associated to the PyG (e.g., if significant changes in PyG happen, the proposed PyTorch-Geometric Edge library might be adjusted very carefully in regards to PyG).
* There is a recent method to include, titled as edge representation learning with hypergraphs [1].

---

## Requested Changes
Note that the below requested changes are related to the weaknesses above.
* Add the discussion on edge direction in the main paper.
* Add the justification of releasing the new repository, instead of using the existing PyG.
* Include the recent edge representation learning method [1].

---

## Questions
* Is the example task in Appendix common? The implemented data processing function, namely MatchingNodeLabelsTransform, additionally generates edge labels based on node labels, thus the edge-level task in the example looks unnatural and unrealistic.

---

[1] Edge Representation Learning with Hypergraphs, NeurIPS 2021.

---

**Changed after author response:** the authors sufficiently address my concerns and suggestions; therefore, I increase my rating from weak accept to clear accept.

---

### Official Review · Reviewer_R8iz · 2022-10-21

**Overall Score:** 8
**Confidence:** 4

**Review:**

This paper introduces Pytorch-Geometric Edge, the first deep learning library for edge-centric representation learning in Graph ML. Specifically, this library provides and evaluates a subset of tasks for link prediction, edge classification, and edge regression. This library re-uses Pytorch Geometric's API and data format.  I think this is a very good starting point for the identified gap in our field, and I hope I will be able to discuss this with the authors at the conference! For the very good strengths I identify down below, and the fact this is a clear fit for the conference, I recommend a clear accept given the subject area (Graph/Geometric ML Infrastructures).

## Strengths:
- This work tackles an overlooked area of Graph ML from public libraries like Pytorch Geometric (PyG) and Deep Graph Library, which are, indeed, mostly focused on node- and graph-centric tasks.
- I think in open source software it is a very good strategy for specific overlooked areas to be developed independently from main libraries so the specific needs can be tailored. A bit like the Unix philosophy of making each program do one thing well. By maintaining most of PyG APIs, this will allow for this library to be merged in the main PyG if a lot of people start successfully using it.
- By maintaining PyG's APIs, it should be easy to "mix" node-centric models and those from this new library, which will probably be a very common case and allow for more use cases.


## Weaknesses and additional feedback:
- The fact that this is still under active development might make it more difficult for people to want to use it. But exactly by being in such early stage of development make it more interesting to be presented in such a conference like LoG, and I'm sure the conference will benefit from this work to be presented and discussed within the audience.
- Unless I misunderstood something, the examples provided show that PyG is a dependency and can be used. In this sense, why does the library provide Cora, PubMed, and KarateClub if they are already provided through PyG, where specific versions are defined for dependencies? It seems that datasets follow exactly the same format as in PyG, so why are the authors re-implementing this in their library?
- Similarly to other platforms, it would be useful to have the roadmap defined in a rough timeframe. This would allow us to see what are the priorities and the timeframes that the authors are thinking as reasonable given the current people coding it.
- There's a typo in line 137, it should have an extra s: "(3) builds".
- I think it would be more correct and realistic to change "ensures" in line 149 to something like "prevents that"
- Given the paper is an extended abstract in itself, I believe there is no need to have an abstract (as the whole paper is an abstract in itself). This could bring some extra space for the authors.

---

### Official Review · Reviewer_mqUo · 2022-10-21

**Overall Score:** 6
**Confidence:** 4

**Review:**

In this extended abstract,  the authors introduced a library PyTorch-Geometric Edge (PyGE), for implementing edge-centric GML models.

The main problem with the work is that the number of layers, datasets and models implemented is still very limited. Further improvements and extensions are also expected.

Whether users can easily modify and develop their own models based on this framework cannot yet be assessed.

Overall it seems to be an encouraging work.

---

### Official Review · Reviewer_poBL · 2022-10-21

**Overall Score:** 8
**Confidence:** 5

**Review:**

*Summary*: The authors implement a Graph Neural Network library on top of PyTorch_Geometric focused primarily on edge representation learning. Additionally, the library also implements edge-based models, dataset and preprocessing related functionalities. This is very useful and much needed functionality for GNNs as majority of the existing libraries only focuses on node features.

*Positives*:
- PyTorch-Geometric Edge will be an asset the in-graph analytics ecosystem. As the majority of existing libraries are node centric (DGL, PyG, etc.) without any support for edge features, this library will ease the implementation burden.
- The library supports multiple edge-base architectures such as LINE, SDNE, or FSCNMF.

*Limitations*:
Although I liked the library, I do have a few queries which I believe should be addressed in the paper.
- Larger Datasets: The largest included dataset has ~20k nodes and ~90k edges, both of which are small compared to real-world graphs. For example, edge classification dataset for intrusion detection (UNSW-NB15) are an order of magnitude larger with ~600k nodes and ~700k edges.
- Another point which I feel the abstract should address, is how this library is different from Knowledge Graph based graph neural network libraries such as DGL-KE?
- It was not clear from the abstract writeup if it supports both node and edge features? I presume it is, but this information should be clearly mentioned.

---

### Meta-Review · Area_Chair_DCaJ · 2022-11-16

**Confidence:** 5
**Recommendation:** Accept

**Meta Review:**

The authors propose a new GNN library for edge representation learning, built upon PyTorch Geometric. The paper is well motivated, since existing GNN libraries focus on node-centric and graph-centric representations. In contrast, the PyTorch-Geometric Edge library provides specific models, datasets, transformations, evaluations and baseline script for edge representation learning missing in existing libraries. This should be highly valuable for the graph ML community.

All reviewers are in consensus of accepting this work. There exists some concerns regarding missing evaluation on larger datasets (with which I agree), so I strongly recommend that the authors include the additional and promised evaluation of larger-scale in the camera-ready version.

---

### Decision · Program_Chairs · 2022-11-23

Accept (Poster)